# Molecular insights into the distinct signaling duration for the peptide-induced PTH1R activation

Xiuwen Zhai [1,8], Chunyou Mao [2,3,8,9] ✉, Qingya Shen [4,8], Shaokun Zang[5], Dan-Dan Shen[5], Huibing Zhang [5], Zhaohong Chen[1], Gang Wang [1], Changming Zhang[1], Yan Zhang [2,4,5,6,7,9] ✉ & Zhihong Liu [1,4,9] ✉

The parathyroid hormone type 1 receptor (PTH1R), a class B1 G protein-coupled receptor, plays critical roles in bone turnover and $Ca^{2+}$ homeostasis. Teriparatide (PTH) and Abaloparatide (ABL) are terms as long-acting and short-acting peptide, respectively, regarding their marked duration distinctions of the downstream signaling. However, the mechanistic details remain obscure. Here, we report the cryo-electron microscopy structures of PTH− and ABL−bound PTH1R-Gs complexes, adapting similar overall conformations yet with notable differences in the receptor ECD regions and the peptide C-terminal portions. 3D variability analysis and site-directed mutagenesis studies uncovered that PTH−bound PTH1R−Gs complexes display less motions and are more tolerant of mutations in affecting the receptor signaling than ABL−bound complexes. Furthermore, we combined the structural analysis and signaling assays to delineate the molecular basis of the differential signaling durations induced by these peptides. Our study deepens the mechanistic understanding of ligand-mediated prolonged or transient signaling.

The parathyroid hormone type 1 receptor (PTH1R) is a prototypical member of class B G protein-coupled receptor (GPCR). PTH1R is highly expressed in bone and kidney where it regulates diverse processes such as skeletal development, bone turnover, mineral ion homeostasis[1]. Aberrant signaling by the defective receptor are known to be the cause of Jansen's metaphyseal chondrodysplasia, Blomstrand's lethal chondroplasia, Ollier's disease and Eiken syndrome, Λ][2−6]. Pharmacologically, PTH1R is the major drug target for the treatment of bone-related diseases such as osteoporosis and the disorders in calcium metabolism[7−9].

PTH1R can be physiologically activated by two distinct endogenous peptide agonists, parathyroid hormone (PTH) and parathyroid hormone-related protein (PTHrP), to regulate multiple signaling pathways and exert distinct biological effects[10]. The gland-secreted PTH regulates calcium and phosphate homeostasis in bone and kidney, while PTHrP modulates cell proliferation and differentiation in developing bone and other tissues[11]. PTH and PTHrP mediate their functions via activating highly similar downstream signaling, but differ significantly in stabilizing distinct PTH1R states and signaling duration[12]. PTH, a long-acting peptide agonist of

[1]National Clinical Research Center of Kidney Diseases, Jinling Clinical Medical College of Nanjing Medical University, Nanjing 211166 Jiangsu, China. [2]Center for Structural Pharmacology and Therapeutics Development, Sir Run Run Shaw Hospital, Zhejiang University School of Medicine, Hangzhou, China. [3]Department of General Surgery, Sir Run Run Shaw Hospital, Zhejiang University School of Medicine, Hangzhou, Zhejiang, China. [4]Liangzhu Laboratory, Zhejiang University Medical Center, Hangzhou, China. [5]Department of Biophysics and Department of Pathology of Sir Run Run Shaw Hospital, Zhejiang University School of Medicine, Hangzhou, China. [6]MOE Frontier Science Center for Brain Research and Brain-Machine Integration, Zhejiang University School of Medicine, Hangzhou, Zhejiang, China. [7]Zhejiang Provincial Key Laboratory of Immunity and Inflammatory diseases, Hangzhou, Zhejiang, China. [8]These authors contributed equally: Xiuwen Zhai, Chunyou Mao, Qingya Shen. [11]These authors jointly supervised this work: Chunyou Mao, Yan Zhang, Zhihong Liu. ✉e-mail: maochunyou@zju.edu.cn; zhang_yan@zju.edu.cn; liuzhihong@zju.edu.cn

PTH1R, efficiently binds to both the G protein-dependent and -independent ($R_G$ and $R_0$) states of PTH1R, inducing prolonged downstream signaling, while PTHrP preferentially binds to the $R_G$ versus $R_0$ state, functioning as a shorting-acting peptide agonist and triggering transient signaling[13,14].

The clinically approved PTH1R-targeted drugs, Teriparatide (human PTH 1-34; hereafter referred to as PTH) and Abaloparatide (an analog of PTHrP; ABL), bear similar properties to the corresponding endogenous ligands[15,16]. Due to the underlying signaling differences, the short-acting ABL shows comparable abilities to stimulate bone formation but displays significantly less concomitant bone resorption and hypercalcaemia when compared with the Teriparatide/PTH[17]. However, the molecular basis for the signaling durations induced by such distinct peptide ligands of PTH1R remains obscure, which hampering further drug optimization and the development of effective, orally available non-peptide agonists for the treatment of osteoporosis and the dysregulations of mineral ion homeostasis.

Herein, we determine the high-resolution cryo-electron microscopy (cryo-EM) structures of human PTH1R-Gs complexes bound to the two FDA-approved drugs, the long-acting PTH and the short-acting ABL. In combination with structural analysis and cellular signaling assays, our work reveals the ligand-binding modes, structural dynamics and the underlying molecular mechanisms of differential signaling durations induced by the distinct peptide agonists. These findings provide significant insights into the activation and downstream signaling of PTH1R by the long- and short-acting peptide agonists, laying the foundation for the development of novel therapeutics for the treatment of related diseases.

## Results

### Pharmacology analysis and structure determination

Consistent with prior studies, our cellular signaling assays showed that ABL and PTH (Fig. 1a) behaved similarly for the signaling profiles of both cAMP accumulation (Fig. 1b) and β-arrestin1 recruitment (Fig. 1c), but exhibited marked differences in the duration of cAMP signaling as measured by "wash-out" experiments (Fig. 1d)[18,19]. PTH displayed potent prolonged cAMP signaling, while ABL exhibited obvious transient cAMP signaling (Fig. 1d). Simultaneously, a long-acting PTH analog (LA-PTH) showed a stronger activity of sustained signaling when compared with the PTH (Fig. 1d). Initially, the attempts to purify ABL and PTH-bound PTH1R-Gs complexes by using our previously established procedure for LA-PTH-bound complex were unsuccessful[20]. LA-PTH, a long-acting PTH analog, exhibited the strongest activity of sustained signaling to our knowledge (Fig. 1d), and not surprisingly stabilized the PTH1R-Gs complex the most among the three peptides. To acquire stable complexes for structural determination, apart from previously used dominant-negative Gαs and the Gs stabilizing nanobody Nb35, we further combined the NanoBiT tethering strategy and used a higher concentration of PTH (50 µM) or ABL

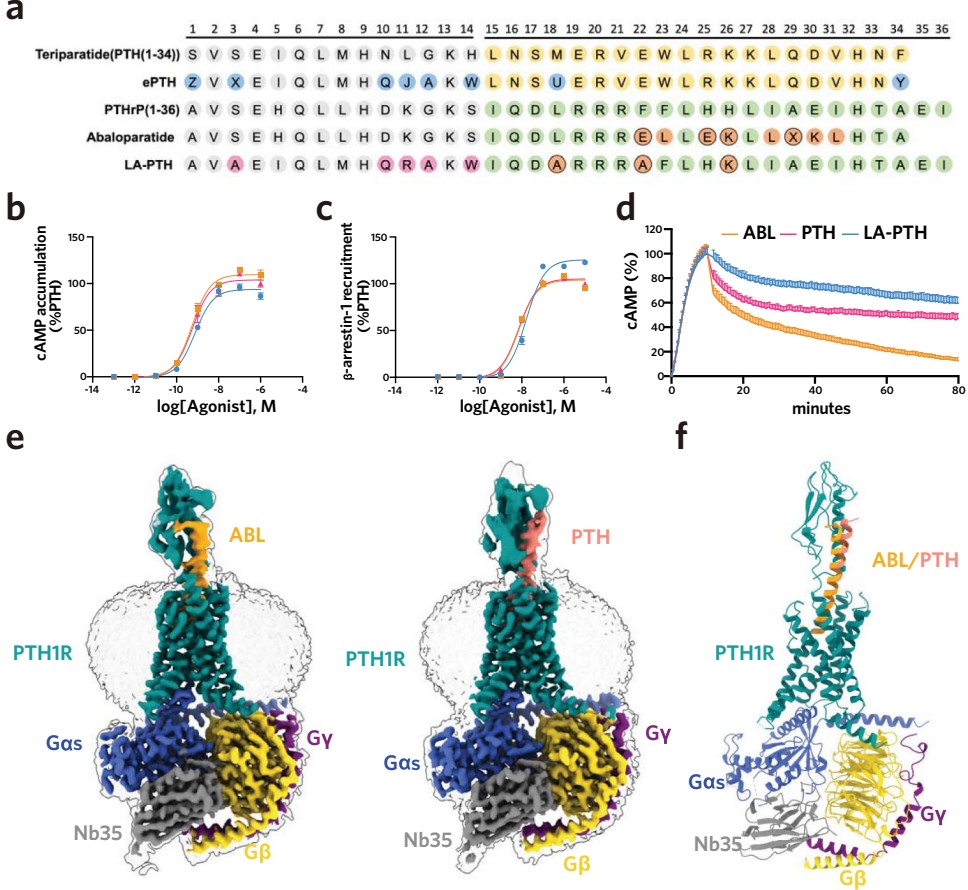

**Fig. 1 | Cryo-EM structures of PTH- and ABL-bound PTH1R-Gs complexes.**
**a** Sequence alignment of the ligands assessed in this study. Relative to PTH, the amino acid substitutions of ePTH are colored blue and the "M" substitutions of LA-PTH are colored pink. Relative to PTHrP, the C-terminal substituted residues of LA-PTH and ABL are colored orange. Amino acids are indicated in one-letter code; Z indicates Ac5c (aminocyclopentane-1carboxylic acid); X indicates Aib (α-methylalanine); J indicates Hrg (homoarginine); U indicates Nle (norleucine). **b, d** Dose-response curves shows the signaling profiles in cAMP accumulation **(b)**, β-arrestin1 recruitment **(c)**, and duration of ligand-induced cAMP signaling responses **(d)**. **e, f** Cryo-EM density maps and cartoon representation of the PTH1R-Gs complexes. PTH, salmon; ABL, orange; PTH1R, dark cyan; Gαs, royal blue; Gβ, yellow; Gγ, purple; Nb35, grey. Source data are provided as a Source Data file.

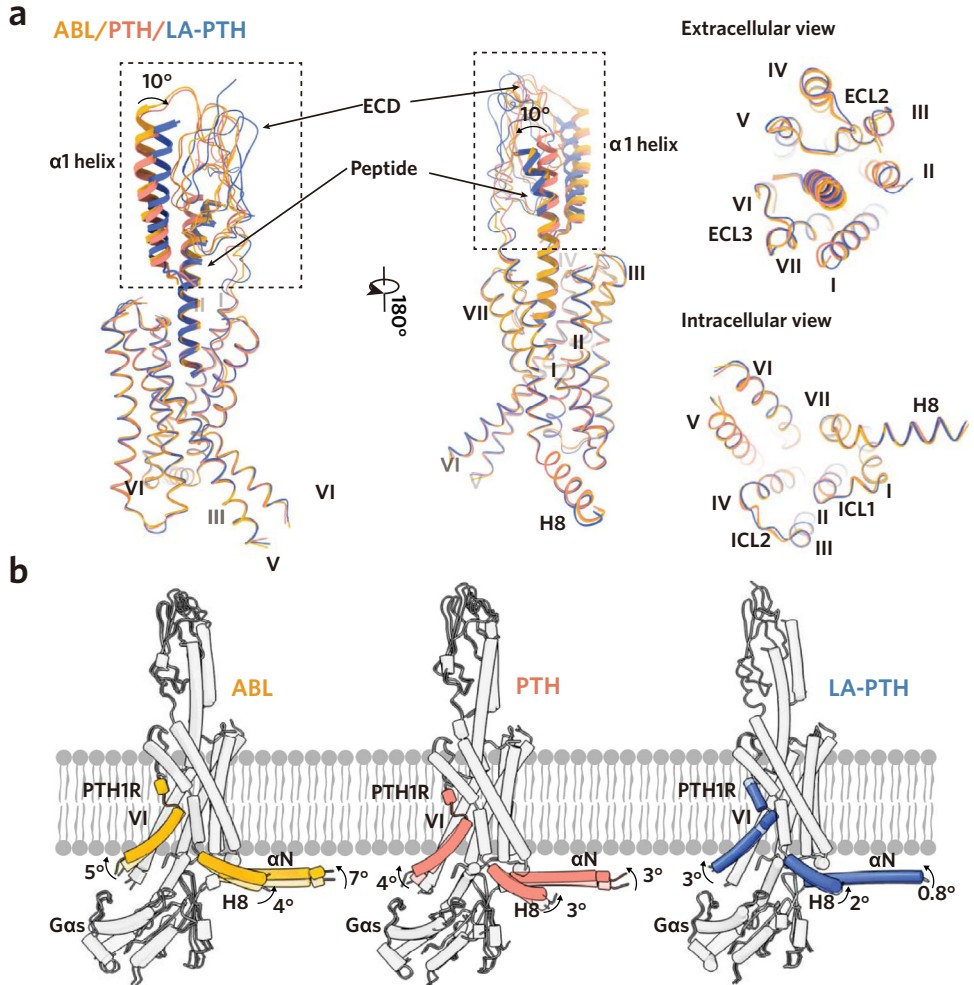

**Fig. 2 | Structural comparisons and dynamics of the peptide-bound PTH1R-Gs complexes. a** Structural comparisons of the ABL−, PTH− and LA-PTH−bound (PDB 6NBF) PTH1R complexes. Structures were aligned by the TMD of PTH1R. **b** 3D-variability analysis of the ABL−, PTH− and LA-PTH−bound PTH1R-Gs complexes. Superimposition of models built into density maps corresponding to frame 000 and frame 019 of components 1 in Supplementary Movies. 1, 2 and 3.

(100 µM) agonists than LA-PTH (10 µM), to assemble the signaling complexes (Supplementary Fig. 1)[21–23].

Cryo-EM data were collected from the vitrified samples of both peptides−bound complexes using a Titan Krios microscope equipped with a K2 detector. After several rounds of 3D classification, we obtained the 3D reconstructions of PTH− and ABL−bound PTH1R-Gs complexes to the nominal global resolutions of 2.8 and 2.9 Å, respectively (Supplementary Figs. 2, 3 and Supplementary Table 1). To further improve the local resolutions of the extracellular domain (ECD) and the seven transmembrane domains (7TMs) of PTH1R, we separately performed local 3D classification and refinement of these regions, which greatly enhances the map quality (Fig. 1e, Supplementary Figs. 2 and 3). Finally, the composite maps from these individual reconstructions were generated for model building (Supplementary Figs. 2 and 3). The high-resolution density maps enable us to build accurate models of both complexes for peptide agonists, the majority residues of Gs as well as the 7TM and most of the extracellular and intracellular loops (ECLs and ICLs) of PTH1R, with the exception of ECL1 and ICL3 (Fig. 1e, f, Supplementary Fig. 4 and Table 1), which were not modeled. Due to the lower resolution of receptor ECD, we docked the high-resolution crystal structure of PTH1R ECD into the density maps as a rigid body, followed by the flexible fitting refinement using Rosetta, which provides a structural framework for subsequent conformational analysis (Fig. 1e, f)[24–26].

## Conformational comparison and dynamics of distinct peptide−bound complexes

Structural comparisons of ABL and PTH−bound complexes with previously determined LA-PTH−bound complex showed that the overall structures of the three peptides−bound complexes are similar, with the Cα root mean square deviation (RMSD) values of 1.7 Å (PTH versus LA-PTH), 1.8 Å (ABL versus LA-PTH) and 1.1 Å (PTH versus ABL), respectively (Supplementary Fig. 5a)[20]. The largest similarities between these complexes occur within the 7TMs of receptor and the coupled Gs protein, with the RMSD values below 1.0 Å (Fig. 2a and Supplementary Fig. 5a). The receptors adopt full active conformations including the sharp kink in the middle of TM6 to accommodate the peptide agonist and the pronounced outward movement at the intracellular end of TM6 to open a cavity for Gs coupling (Fig. 2a and Supplementary Fig. 5a). In addition, the Gs coupling interfaces are almost identical among these structures, mainly formed by ICLs 1/2, TMs 2/3/5/6/7 and helix 8 in the receptor part interacting with the Gαs α5 helix, Gαs β1 loop and Gβ of the Gs protein (Supplementary Fig. 5b−f). The highly similar conformations of the activated PTH1R and Gs coupling interface may be accounted for the comparable efficacy and potency of cAMP accumulation induced by these peptide agonists (Fig. 1b). Nonetheless, we also observed notable differences occurred in the receptor ECD regions between these complexes. Compare with the LA-PTH−bound structure, both the PTH and ABL−bound structures

exhibit a similar 10-degree shift of the ECD relative to the 7TM core (Fig. 2a). Intriguingly, despite this displacement, the distal end of the α1 helix in the ECD domain anchors at similar locations in the middle portions of the peptide agonists (Fig. 2a), indicating that the distal end of the α1 helix probably plays a general role in peptide recognition and stability.

To get insights into the structural dynamics in different peptide complexes, we further performed 3D variability analysis using the final particles for 3D reconstruction. Consistent with the static structural analysis, the individual TMD of the receptor and the Gs heterotrimer were relatively stable, whereas the ECD of these complexes were ambiguous and more dynamic (Fig. 2b, Supplementary Movies. 1, 2, and 3). Intriguingly, 3D variability analysis revealed distinct motions surrounding the Gs interface in different peptide-bound complexes (Fig. 2b and Supplementary Movies. 1, 2, and 3). Specifically, in the ABL-bound complex, the TM6, helix 8 and the coupled Gs exhibited apparent motions, showing a 5-, 4-, and 7-degree rotation relative to the TM core, respectively (Fig. 2b, Supplementary Movie. 1). Meanwhile, similar but relatively smaller motions were observed in the PTH-bound complex, with a 4-, 3-, and 3-degree rotation for the TM6, helix 8 and the coupled Gs, respectively (Fig. 2b and Supplementary Movie. 2). Most strikingly, our previously determined LA-PTH-bound complex displayed marginal motions for these equivalent domains, especially for Gs, which only shows a 0.8-degree rotation (Fig. 2b, Supplementary Movie. 3). Together, these structural observations demonstrated that LA-PTH-bound complex is the most stable, while PTH-bound complex exhibits relatively less stable, whereas the ABL-bound complex shows the most prominent dynamics, which is coincident with the rank order of the signaling duration for these peptide agonists (LA-PTH > PTH > ABL) (Fig. 1d).

### Similarities and differences in the peptide-binding modes

Similar to other peptide ligands of class B1 GPCRs, the peptide agonists of PTH1R are composed of an N-terminal portion that is essential for receptor activation and a C-terminal portion that is important for peptide binding[27]. The sequence homology between the three PTH1R peptide agonists are 45% (PTH versus ABL), 36% (PTH versus LA-PTH), and 57% (ABL versus LA-PTH) (Fig. 1a). Analogous to the previously determined LA-PTH, both PTH and ABL form a continuous α-helix and engage with the TM core and ECD domain of PTH1R (Fig. 3a)[20]. The N-terminal portions of PTH and ABL insert into the TM core, occupying an almost identical central cavity as LA-PTH that is enclosed by TMs 1/2/3/5/6/7 and ECLs 2/3 (Figs. 2a and 3b). Interestingly, the C-terminal portions of PTH and ABL bind to a similar hydrophobic groove in the ECD but with two different conformations (Fig. 3a, b). Similar to LA-PTH, ABL preserves an analogous C-terminal portion of PTHrP and displays a bent α-helix leaning against the short α2 helix of the ECD domain (Fig. 3a, b). By contrast, PTH exhibits a straight α-helix and diverges by approximately 20-degree with ABL and LA-PTH (Fig. 3b). These observations are consistent with the previous crystal structures of PTH1R ECD in complex with the C-terminal portions of PTH and PTHrP[24,25].

To explore the molecular basis of differential signaling behaviors mediated by PTH and ABL, we closely examined the detailed interactions between the two peptides with PTH1R. Both the sequences and side-chain positions of the N-terminal residues (residues 1-14) in PTH and ABL are highly similar, highlighting their great importance in receptor activation (Fig. 1a, Supplementary Fig. 6a). Consistent with this, substitution mutations of most N-terminal peptide residues, particularly residues 1-9, have been noted to substantially reduce downstream cAMP signaling[28]. Sequence analysis showed that there are six amino acid discrepancies within the N-terminal portions of the two peptides (Ser1[PTH]/Ala1[ABL]; Ile5[PTH]/His5[ABL]; Met8[PTH]/Leu8[ABL]; Asn10[PTH]/ Asp10[ABL]; Leu11[PTH]/Lys11[ABL]; His14[PTH]/Ser14[ABL]; superscript indicates the corresponding peptide agonist) (Fig. 1a). Among them, residue divergences in

positions 1, 10 and 11 of PTH and ABL showed no obvious alteration to the detailed interactions (Fig. 3c, Supplementary Tables. 2, 3). Specifically, both the Ser1[PTH] and Ala1[ABL] exhibit extensive polar interactions with the main chain carbonyl group of T427[6.59] and M425[6.57] (class B1 GPCR numbering in superscript)[29] (Fig. 3c, Supplementary Tables. 2, 3). Asn10[PTH] and Asp10[ABL] form hydrogen-bonding interactions with W437[7.35] (Fig. 3c, Supplementary Tables. 2, 3). Leu11[PTH] and Lys11[ABL] are sandwiched similarly between F184[1.36] and Y245[2.72] (Fig. 3c, Supplementary Tables. 2, 3). Nevertheless, we noticed obvious differences occurred in positions 8 and 14. In contrast to Leu8[ABL] that only contacts Y245[2.72], Met8[PTH] forms additional interactions with the residue N353[ECL2] in ECL2 (Fig. 3c, Supplementary Tables. 2, 3). In addition, different from the hydrogen bonding between Ser14[ABL] and E180[1.32], His14[PTH] forms a stronger electrostatic interaction with the identical residue E180[1.32] (Fig. 3c, Supplementary Tables. 2, 3). The most striking distinction is observed in residue position 5. His5[ABL] is primary hydrogen-bonded with Y429[ECL3] and Q364[5.40], whereas Ile5[PTH] forms extensive nonpolar interactions with L292[3.40], L289[3.37] and Q364[5.40] (Fig. 3c, Supplementary Tables. 2, 3).

Relative to the conserved N-terminal portion, the C-terminal portions of the two peptide agonists are strikingly different in the context of protein sequences and structural conformations (Figs. 1a and 3b). Despite the C-terminal half of the two peptides only containing six identical residues (out of twenty) and exhibiting two different conformations, they bind to the same hydrophobic groove in the ECD domain with comparable affinities (Fig. 3a)[30]. The low resolution of the ECD domains limited our ability to clearly define interactions between the C-terminal portion of peptides and ECD domains. However, they were well-characterized previously based on the high-resolution structures of PTH1R ECD domain in complex with the C-terminal portions of PTH and PTHrP[24,25]. Additionally, the critical residues responsible for ECD binding and conformational differences between the two peptides have also been described. Therefore, we do not describe it in detail here[25].

Combined with the LA-PTH-bound structure, we further characterized the detailed interactions between the receptor and the N-terminal portion of the three peptide agonists (Supplementary Fig. 6b). To test whether the distinctions in peptide recognition between these peptides play roles in their signaling profiles, we carried out extensive mutagenesis studies for the residues of the receptor that make differential contacts with the three peptides using cAMP signaling assays (Fig. 3d and Supplementary Figs. 7, 8, 9 and Supplementary Tables. 4, 5, 6). The majority of alanine mutations markedly reduced the potency of ABL, but displayed a moderate or little effect on the potency of PTH and LA-PTH (Fig. 3d and Supplementary Figs. 7, 8, 9 and Supplementary Tables. 4, 5, 6). These results suggest that, despite the similar binding mode and affinities towards $R_G$ state between the short-acting and long-acting peptide agonists, the long-acting peptides induced a more stable signaling complex and are relatively tolerant of receptor mutations when compared with the short-acting peptide.

### Molecular basis for the prolonged and transient cAMP signaling

The central question in the haze of property differences in physiology and pharmacology between PTH and ABL is how they mediate distinct signaling duration, which is strongly pursued owing to its relevance in drug development[31–34]. Previous studies have shown that the long/ short-acting mode for these distinct peptide agonists is highly related to their affinities to $R_G$/$R_0$ conformations of PTH1R[30,35,36]. As mentioned above, the long-acting PTH and LA-PTH could strongly bind to both the $R_G$ and $R_0$ conformations, only showing an approximate 2- to 10-fold differences for binding affinities[18]. By contrast, the short-acting ABL and PTHrP are able to intensively bind to $R_G$ conformation as the long-acting peptides, but show 100- to 1000-folds lower affinities for $R_0$ conformation[18]. This is consistent with our structural observations that

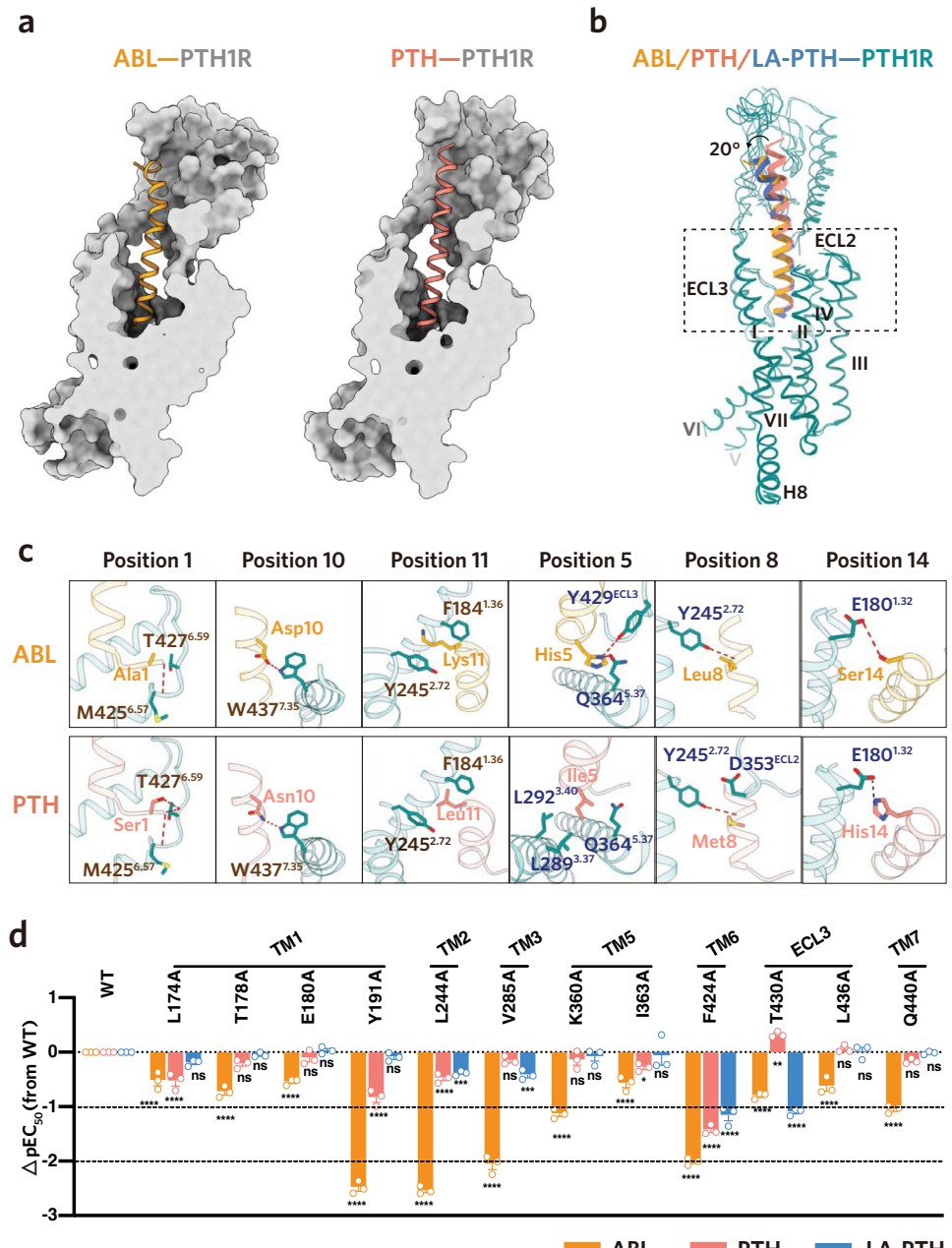

**Fig. 3 | Interactions of PTH and ABL with PTH1R. a** Vertical cross-section of PTH– and ABL–binding pocket in PTH1R. **b** Structural comparisons of the ligand-binding modes between PTH, ABL and LA-PTH. **c** Detailed interactions of PTH1R with the distinct residues in the N-terminal portion of PTH and ABL. Similar interactions are colored in brown (positions 1, 10 and 11), and the differential interactions are colored in blue (positions 5, 8 and 14). Hydrogen bonds and salt bridge are depicted as red and blue dashed lines, respectively. **d** The ABL–, PTH– and LA-PTH–induced cAMP accumulation assays of the PTH1R variants harboring mutations of the residues involved in the differential recognition of three peptide agonists. Bars represent differences in calculated different peptide agonists potency [pEC$_{50}$] for each mutant relative to the wild-type receptor (WT). Data are colored according to the extent of the different peptide agonists. $^{ns}P > 0.01$, $^{*}P < 0.05$, $^{**}P < 0.01$, $^{***}P < 0.001$ and $^{****}P < 0.0001$. All data were analyzed by one-way analysis of variance (ANOVA) followed by Dunnett's multiple comparison test, compared with the response of WT ($P < 0.001$, $P < 0.001$, $P = 0.2164$, $P < 0.001$, $P = 0.1552$, $P = 0.9991$, $P < 0.001$, $P = 0.8638$, $P = 0.9996$, $P < 0.001$, $P < 0.001$, $P = 0.9481$, $P < 0.001$, $P < 0.001$, $P = 0.0006$, $P < 0.001$, $P = 0.4907$, $P = 0.0002$, $P < 0.001$, $P = 0.5859$, $P = 0.9795$, $P < 0.001$, $P = 0.0259$, $P = 0.9991$, $P < 0.001$, $P < 0.001$, $P < 0.001$, $P < 0.001$, $P = 0.0058$, $P < 0.001$, $P < 0.001$, $P = 0.999$, $P = 0.9998$, $P < 0.001$, $P = 0.4148$ and $P = 0.9998$ from left to right). Data represent the mean ± SEM from $n = 3$ biologically independent experiments performed in triplicate. See also Supplementary Figs. 6 and 7. Source data are provided as a Source Data file.

PTH and ABL show a similar binding mode in the PTH1R–Gs signaling complex (Fig. 2a and Supplementary Fig. 5), only displaying differences in complex stability (Figs. 2b and 3d).

Given the great importance of the N-terminal portion of peptide agonists in PTH1R activation and the comparable binding affinities of the C-terminal portions of distinct peptides with the PTH1R ECD[27,30], we speculated that sequence divergences in the N-terminal portions of peptides probably contribute to the distinct signaling behaviors of PTH and ABL. Sequences alignment and detailed structural analysis showed that three residues of the N-terminal portion for PTH and ABL are of great difference containing Ile5[PTH]/His5[ABL], Met8[PTH]/L8[ABL] and His14[PTH]/Ser14[ABL] (Fig. 3c, Supplementary Fig. 10a). To test their potential role, we sought to introduce mutations to the three residues in PTH with the corresponding residues in ABL. Substitution mutations

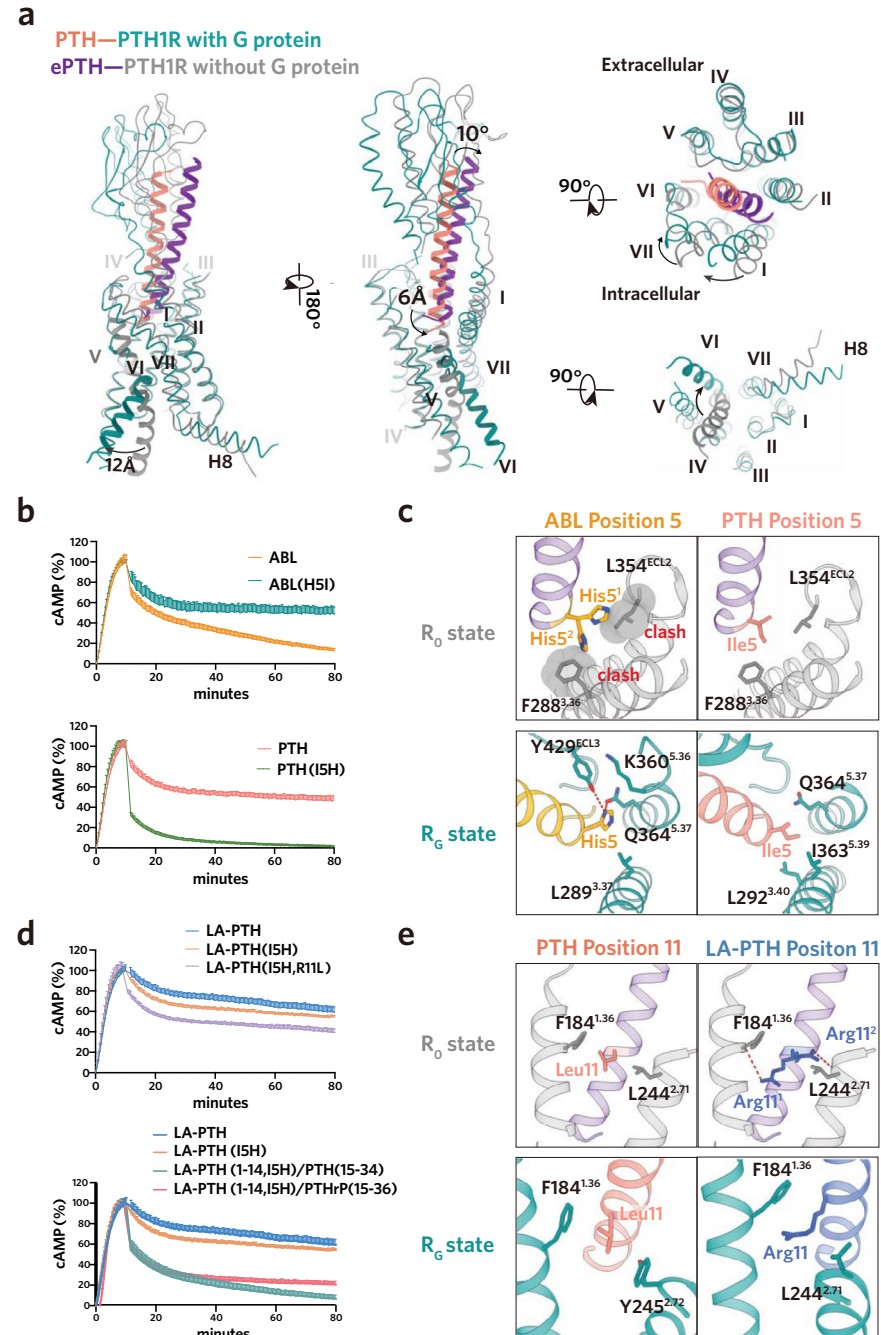

**Fig. 4 | Molecular basis for the prolonged and transient cAMP signaling of PTH1R. a** Structural comparisons of the peptide–bound PTH1R with and without G-protein coupling (ePTH-bound; PDB 6FJ3). **b, d,** Duration of cAMP signaling responses induced by PTH, ABL and LA-PTH analogs. **c, e** Interactions of the peptide residues (positions 5 and 11) with $R_G$ and $R_O$ state PTH1R, respectively. Potential interactions of peptide agonists with $R_O$ state receptor were analyzed based on the template of ePTH-bound PTH1R. (PDB 6FJ3). Source data are provided as a Source Data file.

of residues in positions 8 and 14 exerted no effect on the duration of downstream cAMP signaling examined by "wash out" assay (Supplementary Fig. 10b). Of interest, substituting Ile5 in PTH with the equivalent His5 in ABL abrogated PTH (PTH[I5H])–mediated prolonged cAMP signaling (Fig. 4b). Reciprocally, replacing His5 in ABL with the corresponding Ile5 in PTH renders ABL to induce a sustained cAMP signaling to a comparable extent as PTH (Fig. 4b). These results are in agreement with previous binding experiments showing that the divergent residue in position 5 of PTH and PTHrP (Ile5[PTH]/His5[PTHrP]) plays critical roles in the binding affinity towards $R_O$ but not $R_G$ state[36]. Indeed, our resolved structures confirmed that, despite Ile5[PTH] and His5[ABL] mediated distinct contacts with the G-protein coupled PTH1R

(Ile5[PTH]: hydrophobic interactions; His5[ABL]: polar interactions), the divergent residue 5 showed no alternation to the rotameric state of surrounding residues in PTH1R and the binding mode for the N-terminal portion of peptide agonists in $R_G$ state (Figs. 3c and 4c).

To understand how the divergent residue 5 of the two peptides elicits the differential binding affinity with the $R_O$ state of PTH1R, we compared their potential distinction of the interaction with PTH1R in the absence of G-protein. Previously determined high-resolution crystal structure of human PTH1R in complex with a PTH mimetic agonist (ePTH, sharing 79% sequence identity with PTH) (Fig. 1a) without G protein provides a rational structural template to delineate the peptide binding with the $R_O$ state receptor[37]. The ePTH–bound

PTH1R adopts an inactive-like "intermediate" state with no helix break and sharp kink in TM6 (Fig. 4a) when aligned with the inactive and active structures of class B1 GLP1R and GCGR (Supplementary Fig. 11a, b)[38–41]. Compared to our determined structure of PTH–bound PTH1R–Gs complex ($R_G$ state), substantial conformational differences are observed in the ligand binding pocket of the receptor involving ECD, TMs 1/6/7 and ECLs 2/3 (Fig. 4a). As a consequence, relative to PTH in $R_G$ state, ePTH is rotated by about 10° towards the TM2 and its N-terminal tip sits less deep (-6 Å, measured at the Cα of position 1 residue) to the cavity in TM core (Fig. 4a). A similar phenomenon for the distinctions in the peptide binding modes and the receptor conformations is also observed in the structures of peptide–bound GCGR with and without G-protein (Supplementary Fig. 11c), indicating that this specific peptide agonist–bound state is a representative "intermediate state" prior to G-protein coupling and is probably conserved in class B GPCRs[40,42,43]. Thus, we reasoned that this unique state could be considered as the tentative $R_0$ state. In this $R_0$ state, the Ile5 in ePTH engaged with the hydrophobic residues F288[3.36] and L354[ECL2] (Fig. 4c). Conceivably, the substitution of Ile5 with the equivalent His5 in ABL or PTHrP compromised these hydrophobic contacts. Moreover, this replacement would lead to a potential steric clash between the bulky side chain of His5 with F288[3.36] or L354[ECL2] depending on the rotameric positions of His5 (Fig. 4c). These observations may explain why His5-substitution in ABL results in a weaker binding affinity with $R_0$ state and mediates transient signaling. Interestingly, the sequence variance at residue 5 between PTH and PTHrP also plays an important role in the receptor selectivity between PTH1R and PTH2R[44], determining PTHrP as a weaker agonist on PTH2R (Supplementary Fig. 12).

LA-PTH, a modified PTH/PTHrP hybrid analog (Fig. 1a), exhibits a more potent prolonged cAMP signaling than PTH (Fig. 1d)[45]. Not surprisingly, LA-PTH variant carrying Ile5His replacement only slightly weakened the prolonged cAMP signaling (Fig. 4d). Previous mono-substitutions of the N-terminal portion of PTH (1-14) and the cAMP signaling assays identified several substituted PTH (1-14) analogs with enhanced potency (2- to 10-fold)[28]. A combination of these substitutions (Ala[1,3,12], Gln[10], Arg[11], Trp[14], designated as "M" substitutions) (Supplementary Fig. 13a) exhibited several-fold higher affinity with the $R_0$ state and induced a more potent signaling duration, along with a comparable affinity with $R_G$ state and agonist potency, as compared with PTH[14]. Indeed, our signaling assays showed that the combined mutations of the "M" substitutions in PTH significantly decreased the signaling duration of LA-PTH (Supplementary Fig. 13b). However, the underlying mechanism and the contribution of a specific mono-substitution to the prolonged signaling is lacks of investigation. Detailed analysis of the "M" substitutions in $R_G$ and $R_0$ states suggested that these substitutions had insignificant effects on the detailed contacts with $R_G$ conformation receptor, but exhibited obvious additional interactions with the $R_0$ conformation receptor (Supplementary Fig. 13c). Among these substitutions (Ser1Ala; Ser3Ala; Asn10Gln; Leu11Arg; Gly12Ala; His14Trp), the most noticeable distinctions occurred at Leu11Arg (Fig. 4e). Relative to the original Leu[11] in PTH, the substituted Arg[11] might form potential hydrogen bonds with the side chain of Y245[2.72] and/or the main chain of F184[1.36] (Fig. 4e). Consistent with this observation, our "wash out" assays showed that mutation of Arg[11] to Leu[11] compromised the prolonged signaling in the context of the LA-PTH containing Ile5His mutation (Fig. 4d), but the reversed mutations of the other "M" substitutions showed little effect (Supplementary Fig. 13b). In addition to the "M" substitutions, LA-PTH also incorporated an analog of the C-terminal portion of PTHrP, carrying three residue substitutions (Leu18Ala, Phe22Ala and His26Lys) (Supplementary Fig. 14a) to improve peptide solubility[45]. Previous studies showed that the C-terminal portion of PTHrP bound to PTH1R ECD with a two-fold higher affinity than that of the equivalent part of PTH[30]. Moreover, the interactions of the C-terminal part of the peptide and the receptor ECD are highly conserved in both the $R_G$ and $R_0$ states.

Therefore, the C-terminal portion swapping contributes to the binding with $R_0$ state and enhances the signaling duration, which is confirmed by prior studies and our "wash out" assays (Fig. 4d). Unexpectedly, the reversed mutations of the three residue substitutions in the C-terminal portion of LA-PTH by swapping with the C-terminal portion of PTHrP slightly decreased the signaling duration in the context of the I5H mutation. (Fig. 4d). Close examination of the structures in $R_G$ and $R_0$ states indicated that these residues are located at the solvent-accessible side of the peptide agonist and make little contact with the receptor for both states (Supplementary Fig. 14b). Thus, the increased activity of the prolonged signaling induced by these residue substitutions is probably not resulted from the enhanced binding affinity with the $R_0$ state, but from its optimized physicochemical property. Collectively, our results, alongside prior studies[14,18,20,25,36,37,45], underlie the molecular basis of the differential affinities toward the $R_G$ and $R_0$ states and the distinct signaling duration by the long-/short-acting peptide agonists, providing a comprehensive understanding of the activation and signaling of PTH1R.

## Discussion

PTH1R has been considered as a major mediator of skeletal development, bone turnover and calcium homeostasis in bone and kidney and a validated drug target for osteoporosis[46]. Physiologically, PTH1R can be activated by two distinct peptide agonists, PTH and PTHrP, which exhibit similar downstream signaling pathways but differ apparently in the signaling durations (Fig. 1)[10]. Similar to the endogenous agonists, two analogs have been developed for the treatment of osteoporosis. PTH (referring to Teriparatide) is the first-in-class drug that can stimulate bone formation by targeting PTH1R[47], but with the accompanying side effects of bone resorption and hypercalcemic. The short-acting ABL, a second-generation osteoanabolic drug, shows a similar capacity for bone formation with fewer side effects than PTH (Teriparatide)[31]. Besides, a more potent long-acting peptide LA-PTH than PTH has also been developed as a potential drug for the treatment of hyperthyroidism by producing sustained calcemic responses[48]. The pharmacological distinctions of these peptides are probably resulted from their distinct capacities to induce the durations of downstream signaling. However, the underlying mechanisms are not fully understood.

In this study, we determined the cryo-EM structures of PTH and ABL–bound PTH1R–Gs complexes (Fig. 1e, f). Combined with previously determined LA-PTH–bound structure[20], we demonstrated the conformation and detailed structural information of the three distinct peptides–bound PTH1R complexes in $R_G$ state. In line with their similar affinities to $R_G$ state receptor and the comparable potency and efficacy of the cAMP signaling profiles, PTH, ABL and LA-PTH showed high similarities in the ligand-binding mode, receptor conformation and the G-protein coupling interface (Figs. 2a, 3 and Supplementary Fig. 5). Nonetheless, 3D variability analysis and mutagenesis studies revealed that the three peptides–bound complexes differed clearly in the complex stability (Figs. 2b and 3d). The most potent long-acting LA-PTH exhibited the least motions, while the short-acting ABL displayed the most significant structural dynamics. The rank order (LA-PTH > PTH > ABL) for the complex stability is consistent with the signaling duration of these peptides, which may partially explain the mechanisms of their distinct signaling duration. In addition to $R_G$ state, we also tried to characterize the distinctions of the potential interactions of these peptides with the receptor in $R_0$ state based on the reported structure of ePTH–bound PTH1R without G-protein[37]. The ePTH–bound PTH1R resembled the glucagon analogue–bound GCGR without G-protein for both the peptide-binding mode and the receptor conformation but significantly differed from their corresponding $R_G$ states, indicating that the unique conformation of the peptide–bound receptor complex before G-protein coupling is a conserved "intermediate state" in class B GPCR (Fig. 4a and Supplementary

Fig. 11c)[40,42,43]. Therefore, it is reasonable to take it as a representative $R_0$ state. Combined the structural analysis and signaling assays, we further identified the critical residue (I/H in position 5) that differentiate the distinct signaling duration between PTH and ABL, due to its alternation to the affinities with the $R_0$ state receptor R (Fig. 4b, c). Regarding LA-PTH, its long-acting property was further enhanced by the N-terminal "M" substitutions (especially L11R replacement) and the C-terminal portion swapping with PTHrP (Fig. 4d, e). Both the replacements contribute to the high binding affinities of peptide with the $R_0$ state. Collectively, our structural and functional findings uncovered the molecular basis for the distinct signaling duration of these peptide agonists, which provides in-depth understanding of the PTH1R signaling. However, further investigations are required to depict a more comprehensive understanding of the molecular mechanisms underlining the differential signaling durations induced by distinct peptide agonists, including the structure determination of PTH- and PTHrP–bound PTH1R in $R_0$ state and extensive peptide binding assays with both the $R_0$ and $R_G$ state receptor.

Based on our results and prior studies, we proposed that the long-acting peptide such as PTH and LA-PTH could strongly induce and stabilize the "intermediate state" when engaged with the inactive receptor due to its high affinity to the $R_0$ state. Subsequently, the peptide–bound "intermediate state" drives the conformational changes of the TMs, particularly TM6, to fully accommodate the peptide into the TM core and couple downstream Gs protein, resulting in the ultimate transitions to the more stable fully active state[43]. Given its high affinity to both states (intermediate and G-protein coupling states), the long-acting peptide maintains a ligand–bound state for multiple cycles of G protein coupling and dissociation even the complex has moved from the cell surface into the endosomal compartments[49], thus resulting in sustained downstream signaling. By contrast, the short-acting peptide easily escapes from the receptor during the activation cycles because of its incompatible binding with "intermediate state" and its unstable $R_G$ state, therefore leading to transient signaling[49]. Given the relatively similar activation mechanisms of class B1 receptors[50], the mechanism might be applicable to other receptors, pointing the way toward the development of the long- or short-acting ligands for class B1 GPCRs.

# Methods

## Constructs

The wild-type (WT) human PTH1R was cloned into a modified pFastBac vector with two mutations (G188A and K484R), which did not affect the receptor ligand binding or activation[20]. The native signal peptide was replaced with the haemagglutinin signal peptide (HA) to enhance receptor expression. The primers used in this study are shown in Supplementary Table 7. To facilitate expression and purification, LgBiT and a double MBP tag were fused to the C-terminus with a TEV protease cleavage site between them. Dominant negative $G\alpha_s$ (DNG$\alpha_s$) includes mutations G226A and A366S of $G\alpha_s$ to reduce nucleotide binding affinities and enhance the stability of PTH1R and $G\alpha_s$ heterotrimer complex[21,51]. G$\beta$1 was fused with an N-terminal HiBiT and 10x His tag, and was cloned into pFastBac dual vector together with G$\gamma$2. For cAMP accumulation, wash out, and ELISA assay, a flag-tag was fused to the N terminal of full-length PTH1R (WT or mutant), and cloned into pcDNA3.1 plasmids. For $\beta$-arrestin1 recruitment assay, the LgBiT was inserted into the C terminal of PTH1R, and the SmBiT was N terminally fused to $\beta$-arrestin1. These sequences were then cloned into pBiT1.1 plasmids (Promega). All the constructs were confirmed by sequencing.

## Expression and purification of Nb35

Nb35 with a C-terminal 6x His tag was expressed in the periplasm of *Escherichia coli* WK6 cells. The cells containing the recombinant plasmid were cultured in TB medium supplemented with 0.1% glucose, 1 mM MgCl$_2$ and 50 μg/mL kanamycin at 37 °C until OD$_{600}$ reached 0.6.

Then the cultures were induced by 1 mM IPTG and grown at 18 °C for 24 h. The cells were harvested by centrifugation at $4000 \times g$ for 15 min, and subsequently lysed. Proteins were purified by nickel affinity chromatography[52]. Eluted proteins were concentrated using a 10 kDa molecular weight cut-off concentrator (Millipore) and loaded onto a HiLoad 16/600 Superdex 75 column (GE Healthcare) with running buffer containing 20 mM HEPES pH 7.5 and 100 mM NaCl. The monomeric fractions were pooled. Purified Nb35 was finally flash frozen in liquid nitrogen and stored at −80 °C.

## Complex formation and purification

Sf9 insect Cell cultures were grown in protein-free insect cell culture medium (Expression Systems ESF 921, Cat. 94011 S) to a density of $2.2 \times 10^6$ cells per mL and then infected with three separate virus preparations for PTH1R, DNG$\alpha$s and G$\beta$1$\gamma$2 at equal MOIs. The infected cells were cultured at 27 °C for 48 h before collection by centrifugation and the cell pellets were stored at −80 °C.

The cell pellets were thawed on ice and lysed in a buffer containing 20 mM HEPES pH 7.5, 150 mM NaCl and 2 mM MgCl$_2$ supplemented with EDTA-free protease inhibitor cocktail (Bimake) by dounce homogenization. The complex formation was initiated by addition of 10 μg/mL Nb35, 25 mU/mL apyrase (NEB) and adequate agonist (50 μM PTH (1-34) or 100 μM Abaloparatide; Synpeptide). The cell lysate was subsequently incubated for 1 h at room temperature (RT) and then solubilized by 0.5% (w/v) lauryl maltose neopentyl glycol (LMNG, Anatrace) and 0.1% (w/v) cholesterol hemisuccinate (CHS, Anatrace) for 2 h at 4 °C. After centrifugation at $30,000 \times g$ for 30 min, the supernatant was isolated and incubated with amylose resin (NEB) for 1 h at 4 °C. Then the resin was collected by centrifugation at $600 \times g$ for 10 min and loaded into a gravity flow column (Beyotime), and washed with five column volumes (CVs) of buffer containing 20 mM HEPES pH 7.5, 150 mM NaCl, 2 mM MgCl$_2$, agonist (5 μM PTH (1-34) or 10 μM Abaloparatide), 0.01% (w/v) LMNG and 0.005% (w/v) CHS, eluted with 15 CVs of buffer containing 20 mM HEPES pH 7.5, 150 mM NaCl, 2 mM MgCl$_2$, agonist (5 μM PTH (1-34) or 10 μM Abaloparatide), 0.01% (w/v) LMNG, 0.005% (w/v) CHS and 10 mM maltose. The elution was collected and incubated with TEV protease for 1 h at RT. Then the elution was concentrated with a 100 kDa cut-off concentrator (Millipore). Concentrated complex was loaded onto a Superose 6 increase 10/300 GL column (GE Healthcare) with running buffer containing 20 mM HEPES pH 7.5, 150 mM NaCl, 2 mM MgCl$_2$, agonist (5 μM PTH (1-34) or 10 μM Abaloparatide), 0.00075% (w/v) LMNG, 0.0002% (w/v) CHS and 0.00025% (w/v) GDN (Anatrace). The fractions for the monomeric complex were collected and concentrated for electron microscopy experiments.

## Cryo-EM grid preparation and data collection

Three microlitres of the purified complexes at approximately 30 mg ml$^{-1}$ were applied onto the glow-discharged holey carbon grids (Quantifoil, R1.2/1.3, 300 mesh). The grids were blotted for 3.5 s with a blot force of 5 at 4 °C, 100% humidity, and then plunge-frozen in liquid ethane using Vitrobot Mark IV (Thermo Fischer Scientific). Cryo-EM data collection was performed on a Titan Krios at 300 kV accelerating voltage in the Center of Cryo-Electron Microscopy (Zhejiang University). Micrographs were recorded using a Gatan K2 Summit Detector in super-resolution mode with a pixel size of 1.014 Å using SerialEM software[53]. Image stacks were obtained at a dose rate of 8.0 e/Å$^2$/s with a defocus ranging from −1.0 to −2.5 μm. The total exposure time was 8 s, and 40 frames were recorded per micrograph. A total of 6947 and 3623 movies were collected for the PTH and ABL–bound complexes, respectively.

## Cryo-EM data processing

Image stacks were aligned using MotionCor 2.1[54]. Contrast transfer function (CTF) parameters were estimated by Gctf v1.18[55]. The following data processing was performed using RELION 3.1[56].

For PTH–bound complex, automated particle selection using Gaussian blob detection in RELION produced 3,607,078 particles. The map of LA-PTH–bound PTH1R–Gs complex[20] (EMD-0410) low-pass filtered to 40 Å was used as the initial reference map. The picked particles were subjected to two rounds of 3D classifications to discard bad particles, resulting in 1,343,480 particles. Further 2 rounds of local 3D classifications focusing on the alignment of micelle-subtracted complex or the receptor in RELION generated two well-defined subsets with 922,971 particles. To further improve the local resolution of the complex, another round of focused 3D classification on the TMG (the transmembrane domains of the receptor and G protein) or ECD was performed. The produced good subsets were subsequently subjected to a final round of 3D refinement, CTF refinement and Bayesian polishing, generating the final maps with indicated global resolutions of 2.6 and 3.2 Å, respectively. The final refinement map was sharpened with deepEMhancer[57] and combined used for subsequent model building and analysis.

For ABL–bound complex, 2,036,564 particles generated from the template-based particle picking were subjected to 2 rounds of 3D classification using RELION (v3.1). Then the good subset accounting for 666,920 particles was refined and subjected to local 3D classification focusing on the receptor, producing one high-quality subset accounting for 456,840 particles. To further improve map quality, another round of focused 3D classification on the TMG or ECD was performed as the Teriparatide–bound complex. The final good particles were subjected to 3D refinement, CTF refinement and Bayesian polishing, generating the final maps with indicated global resolutions of 2.8 and 3.1 Å, respectively. The final refinement map was sharpened with deepEMhancer[57] and combined used for subsequent model building and analysis. Local resolution was determined using the Bsoft package[58] with half maps as input maps.

3D variability analysis of ABL–, PTH–, and LA-PTH–bound PTH1R-Gs complexes were performed in CryoSPARC (v3.2) with 2 components mode[59]. For LA-PTH-bound complex, the final 3D reconstruction data from our previous study was used[20]. Then, the model of each peptide–bound complex was subjected to flexible fitting into the corresponding two maps (frame000 and frame019) of component 1 and 2 using Rosetta 2019.35[26]. Movies were generated using UCSF ChimeraX[60].

## Model building and refinement

The structure of LA-PTH–bound PTH1R–Gs complex (PDB: 6NBF)[20] was used to generate the initial template. Models were docked into final density map using UCSF Chimera[61] (v1.14). The docked models were subjected to flexible fitting using Rosetta[26] (v2019.35) and were further rebuilt in Coot[26] (v0.8.9) and real-space-refined in Phenix[62] (v1.18). The final refinement statistics were validated using the module 'comprehensive validation (cryo-EM)' in Phenix. The goodness-of-fit of the models to the maps were determined using a global model-versus-map Fourier shell correlation. The refinement statistics are provided in Supplementary Table. 1. Structural figures were created using UCSF Chimera and the UCSF Chimera X[60] package.

## cAMP accumulation assay

Human embryonic kidney 293 T (HEK293T, ATCC: CRL-1573) cells were transfected with a plasmid mixture consisting of pcDNA3.1-flag-PTH1R (WT or mutants) and the cAMP biosensor GloSensor-22F (Promega) at a ratio of 2:1 (see Supplementary Table 7 for a list of primers used in this study). After 24 h, transfected cells were plated onto a 384-well plate, which was treated with cell adherent reagent (Applygen) in advance. After another 12 h, cells were treated with Hank's balanced salt solution for starvation and then incubated in $CO_2$-independent media containing 4% GloSensor cAMP Reagent (Promega) at a volume of 5 μl per well and incubated for 20 min at RT before measurements for baseline luminescence (Spark Multimode microplate reader,

TECAN). Next, test ligands were added at different concentrations from $10^{-6}$ to $10^{-13}$ M before second measurement. Finally, data were normalized to 100% of the maximal PTH (1-34) response for WT PTH1R using a sigmoidal dose-response in GraphPad Prism.

## Wash out assay

Human embryonic kidney 293 T (HEK293T) cells were transfected with a plasmid mixture consisting of pcDNA3.1-flag-PTH1R (WT) and the cAMP biosensor GloSensor-22F (Promega) at a ratio of 2:1. After 24 h, transfected cells were plated onto a 96-well plate, which was treated with cell adherent reagent (Applygen) in advance. After another 12 h, cells were treated with Hank's balanced salt solution for starvation and then incubated in $CO_2$-independent media containing 4% GloSensor cAMP Reagent (Promega) at a volume of 40 μl per well and incubated for 20 min at RT before measurements for baseline luminescence (Spark Multimode microplate reader, TECAN). Next, test ligands (10 μl) were added at $10^{-6}$ M that approximated the respective Emax (efficacy) value obtained for that "ligand- on" phase were similar for the different ligands. Then the development of cAMP-dependent luminescence was measured for 10 min ("ligand-on" phase); the plates were then removed from the plate reader, the cells were rinsed twice to remove unbound ligand, treated with fresh media containing luciferin, and luminescence was again recorded for an additional 70 min ("wash-out" phase)[18]. Finally, data were analyzed using GraphPad Prism.

## Cell-surface ELISA

The transfected cells were washed with 1x PBS and fixed with 4% paraformaldehyde for 10 min. Following fixation, cells were blocked with blocking buffer (1% (w/v) BSA/PBS) for 1 h at RT. Afterward, cells were incubated with a 1:10,000 dilution of anti-FLAG M2 HRP-conjugated monoclonal antibody (Sigma-Aldrich, Catalog Number A8592, Mouse lgG1) in blocking buffer for another 0.5 h at RT. Then, wells were washed with three times blocking buffer and three times 1x PBS in order. Finally, antibody binding was detected using 80 μL/well diluent SuperSignal Elisa Femto Maximum Sensitivity Substrate (Thermo-Fisher Scientific). The plate was measured for luminescence (Spark Multimode microplate reader, TECAN). Finally, data were normalized to 100% of the WT PTH1R using GraphPad Prism.

## β-arrestin1 recruitment assay

The NanoBiT assay for the measurement of β-arrestin1 recruitment was performed[63]. Human embryonic kidney 293 T (HEK293T) cells were transfected with a plasmid mixture consisting of pBiT1.1-flag-PTH1R (WT) and the pBiT1.1-β-arrestin1 at a ratio of 1:3 (see Supplementary Table 7 for a list of primers used in this study). After 24 h, transfected cells were plated onto a 96-well plate, which was treated with cell adherent reagent (Applygen) in advance. After another 12 h of incubation, the transfected cells were washed once with Hank's balanced salt solution buffer, and then maintained in the same buffer containing 5 mM HEPES pH 7.5, 0.01% BSA and 5 μM coelenterazine h (yeasen) at a volume of 25 μl per well. After incubation for 30 min, the plate was measured for baseline luminescence (Spark Multimode microplate reader, TECAN). Test ligands (5 μl) were added at different concentrations from $10^{-6}$ to $10^{-13}$ M before second measurement. Finally, data were normalized to 100% of the maximal PTH (1-34) response for WT PTH1R using a sigmoidal dose response in GraphPad Prism.

## Statistical analysis

Statistical analyses were performed on at least three individual datasets and analysed using GraphPad Prism software. Bars represent differences in the calculated agonist potency (pEC$_{50}$) and maximum agonist response (E$_{max}$) for each mutant relative to the WT receptor. Data are mean± s.e.m. from at least three independent experiments, performed in triplicates. ns, not significant, $^{ns}P > 0.01$; $^*P < 0.01$; $^{**}P < 0.001$; $^{***}P < 0.0001$ (one-way analysis of variance (ANOVA)

**Article**

followed by Dunnett's test, compared with the response of the WT). For dose-response experiments, data were normalized and analyzed using nonlinear curve fitting for the log (agonist) versus response (three parameters) curves.

## Peptide synthesis

All peptides used in this study were synthesized in Synpeptide (Nanjing, China). The biophysical analyses and purity assessment of these peptides are provided in source data. The sequences for these peptide ligands are listed in the Source Data Table.1.

## Reporting summary

Further information on research design is available in the Nature Research Reporting Summary linked to this article.

## Data availability

The data that support this study are available from the corresponding authors upon reasonable request. The cryo-EM maps have been deposited in the Electron Microscopy Data Bank (EMDB) under accession codes 33588 (ABL-bound PTH1R-Gs complex) and 33590 (PTH- bound PTH1R-Gs complex). The atomic coordinates have been deposited in the Protein Data Bank (PDB) under accession codes 7Y35 (ABL-bound PTH1R-Gs complex) and 7Y36 (PTH- bound PTH1R-Gs complex). The structural data used in this study were retrieved from the PDB using accession codes 6NBF (LA-PTH–PTH1R–Gs complex), 6FJ3 (ePTH–PTH1R complex), 6LMK (Glucagon–GCGR–Gs complex), 5XEZ (inactive GCGR), 6VCB (GLP-1GLP-1R–Gs complex), 6LN2 (inactive GLP-1R), 5YQZ (Glucagon analogue–GCGR complex), and 7F16] (TIP39–PTH2R–Gs complex). Source data underlying Figs. 1b–d, 3d, 4b, d; Supplementary Figures 1a, c, 7–10, 12; and Supplementary Table 4–6 are available as a Source Data file. Source data are provided with this paper.

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

## Acknowledgements

The cryo-EM data were collected at the Cryo-Electron Microscopy Center, Zhejiang University with technical support from S. Chang. Protein purification was performed at the Protein Facilities, Zhejiang University School of Medicine. This project was partially supported by the Special Funds of the National Natural Science Foundation of China (32141004 to Z.L.), National Key Research and Development Project of China (2021YFC2501302), the National Natural Science Foundation of China (81922071 to Y.Z, 32100959 to C.M., 82070755 to Z.C. and 82170739 to C.Z.), the Zhejiang Provincial Natural Science Foundation of China (LR19H310001 to Y.Z., LR22C050002 to C.M. and LR19H100001 to C.Z.), the Jiangsu Province Natural Science Foundation of China (BK20210150 to G.W.), the Ministry of Science and Technology (2019YFA050880 to Y.Z.), and the Key R&D Projects of Zhejiang Province (2021C03039 to Y.Z.). Y.Z. is also supported by the Fundamental Research Funds for the Central Universities.

## Author contributions

Z.L. and Y.Z. conceived and supervised the project. X.Z. designed the constructs, expressed and purified the PTH1R–Gs complexes; D.-D.S. evaluated the sample by negative-stain EM; C.M. prepared the cryo-EM grids, collected the cryo-EM data and performed cryo-EM map calculation and model building; X.Z. generated the constructs of mutants and performed the cellular functional assays; Z.L., Y.Z., C.M., X.Z., and Q.S. analyzed the data; X.Z., and Q.S. prepared the figures; S.Z., H.Z., Z.C., C.Z., and G.W. participated in data analysis and manuscript preparation. Z.L., Y.Z. and C.M. wrote the manuscript with inputs from all the authors.

## Competing interests
The authors declare no competing interests.
