## [Peer Review File · Nature Communications]

Molecular insights into the distinct signaling duration for the peptide-induced PTH1R activationReviewers' Comments:

Reviewer #1:

Remarks to the Author:

PTH1R plays a vital role in bone turnover and Ca²⁺ homeostasis and is a major drug target for osteoporosis. Of interest, PTH1R can be activated by two endogenous peptide agonists, PTH and PTHrP, displaying distinct physiological or pharmacological properties. In this manuscript, Zhai et al. reported the cryo-EM structures of PTH1R in complex with Gs and PTH as well as PTHrP analog Abaloparatide. The structure is of high quality to reveal molecular recognition and conformational dynamics for the short- and long-acting peptides-bound PTH1R. In addition, this study conducted massive structural and functional analyses to characterize the differential signaling durations induced by these peptides. The structural data is impressive and the experimental data is supportive, providing mechanistic insights into the ligand recognition and the distinct signaling duration of PTH1R. The figures are on the whole very clear and support the text. I think this is a high-quality study and the manuscript is well-written. I would like to recommend accept this manuscript after following minor point addressed.

(1) ABL, PTH and LA-PTH behaved similarly for the signaling profiles of cAMP accumulation, but exhibited marked differences in the duration of cAMP signaling. Some novel peptides designed in this manuscript reversed the pattern of signaling duration for PTH and ABL. The author should measure the dose-response curves of the novel peptides in Figure 4 and determine the EC₅₀ and E_{max}.

(2) In contrast to PTH that activates both PTH1R and PTH2R, PTHrP only has a weak action on PTH2R. The authors could compare the structures of the peptide-bound PTH1R and PTH2R and discuss the potential mechanism.

(3) In Figure 3c, the dash lines between the receptor and ligand are shown in two colors. The author should add clear descriptions for them in the figure legend.

(4) TMG in Supplementary Table.1, Supplementary Figs. 2b and 3b should be expanded in full name when firstly mentioned.

(5) The version of computational programs including RELION, cryoSPARC and Rosetta should be included.

Reviewer #2:

Remarks to the Author:

In the manuscript titled "Molecular insights into the distinct signaling duration for the peptide-induced PTH1R activation", the authors present two cryo-electron microscopy structures of PTH1R bound by Teriparatide (PTH) or Abaloparatide (ABL) as well as comprehensive functional studies to reveal the underlying mechanisms for peptides recognition and their mediated prolonged or transient signaling. The work is of high quality and the article is written well. It has the potential to add to the area of GPCR pharmacology, especially for class B1 GPCRs. There are few issues that should be addressed by the authors, and there are errors that must be corrected.

Major points

1. As demonstrated in the paper, PTH and PTHrP are two endogenous agonists that activating highly similar downstream signaling, but differ significantly in stabilizing distinct PTH1R states and signaling duration. The molecular determinations of the peptide-mediated signaling dynamics are of critical importance for PTH1R-targeted drug discovery, and this manuscript present significant efforts on this direction. Is there some technological problems on determining PTHrP-bound structures? Alternatively, further molecular dynamic simulation of PTHrP-bound PTH1R based on ABL-PTH1R structure could strengthen current efforts. Where, I'm not surprising to see the sequence variance at the peptide is involved in their distinct signaling duration.

2. The hypothesis of peptide-binding affinity in different receptor states (R_o and R_G) that may correlate with the long/short acting and/or prolonged/transient signaling is very interesting and

impressive. Such hypothesis could be further supported by the peptide-binding assay in Ro or RG states as well as the G protein binding. Is there any evidence or experimental data (especially for the new-designed peptide mutants) could be performed? If not, further discussion is encouraged.

Minor points

1. The orders of residue positions (1, 10, 11, 5, 8, 14) in Fig. 3c is kind of unusual, and the addition of LA-PTH in such binding pose comparison could provide further information. Besides, as there are three cryo-EM structures of LA-PTH-PTH1R-Gs complexes, the detailed PDB code should be mentioned.
2. Fig.4 legend: Position 5 and 11 \diamond positions 5 and 11.
3. Supplementary Fig.5a. The colors of ABL- and PTH-bound PTH1R are close.
4. The source of the ligands (ABL and PTH) should be indicated and the biophysical analyses and purity assessment of the peptides should be reported.

Responses to the reviewers' comments

We thank the referees for their valuable time reviewing our manuscript and the constructive suggestions that they have provided. We have carefully taken these comments into consideration when preparing a revision, which resulted in a more thorough and clear manuscript. We have copied each comment in **Black Italic**, which is followed by our own point-by-point response in **Blue**, including details about the corresponding changes to the manuscript.

Reviewer #1

PTH1R plays a vital role in bone turnover and Ca²⁺ homeostasis and is a major drug target for osteoporosis. Of interest, PTH1R can be activated by two endogenous peptide agonists, PTH and PTHrP, displaying distinct physiological or pharmacological properties. In this manuscript, Zhai et al. reported the cryo-EM structures of PTH1R in complex with Gs and PTH as well as PTHrP analog Abaloparatide. The structure is of high quality to reveal molecular recognition and conformational dynamics for the short- and long-acting peptides-bound PTH1R. In addition, this study conducted massive structural and functional analyses to characterize the differential signaling durations induced by these peptides. The structural data is impressive and the experimental data is supportive, providing mechanistic insights into the ligand recognition and the distinct signaling duration of PTH1R. The figures are on the whole very clear and support the text. I think this is a high-quality study and the manuscript is well-written.

Response: We thank the reviewer for the positive comments on our manuscript.

I would like to recommend accept this manuscript after following minor point addressed.

(1) ABL, PTH and LA-PTH behaved similarly for the signaling profiles of cAMP accumulation, but exhibited marked differences in the duration of cAMP signaling. Some novel peptides designed in this manuscript reversed the pattern of signaling duration for PTH and ABL. The author should measure the dose-response curves of the novel peptides in Figure 4 and determine the EC₅₀ and E_{max}.

Response: We appreciate the referee's valuable comments. We have measured the dose-response curves of all 16 new-designed peptide ligands used in this manuscript and determined their EC₅₀ and E_{max} values. These results are shown below and the raw data are also incorporated into the **Source data**.

Source data Figure.1 | cAMP accumulation assays of the new-designed peptide ligands used in this study.

Dose-response curves of cAMP accumulation were measured by Glosensor assay. The data were analyzed using the ‘log(agonist) vs. response-Variable slope (four parameters)’ function in Graphpad Prism. All data are presented as mean values \pm standard error of measurement (SEM).

Source data Table .2 | cAMP dose-response analyses in HEK293T cells^a

	pEC50	Emax
PTH	-8.448	103.1
LAPTH(I5H)	-9.738	120.8

LAPTH(1-14,I5H)/PTHrP(15-34)	-8.755	142.8
LAPTH(1-14,I5H)/PTH(15-34)	-9.192	129.6
LA-PTH (I5H-PTH;NT)	-8.703	102.1
LAPTH(I5H;A1S)	-8.735	119.8
LAPTH(I5H;A3S)	-9.409	125.3
LAPTH(I5H;Q10N)	-8.854	117.7
LAPTH(I5H;R11L)	-8.682	106.9
LAPTH(I5H;A12G)	-8.681	129.5
LAPTH(I5H;W14H)	-9.307	117.8
LAPTH(I5H)	-9.738	120.8
LAPTH(M8L)	-8.462	99.71
LAPTH(W14H)	-8.974	107.5
ABL(H5I)	-9.495	104.1
PTH(I5H)	-7.628	108.9

a, Assays were performed in HEK293T cells; values of half-maximal stimulatory concentration (as pEC50) and Emax (as luminescence counts per 500ms) were derived from curve fitting dose-response data.

(2) *In contrast to PTH that activates both PTH1R and PTH2R, PTHrP only has a weak action on PTH2R. The authors could compare the structures of the peptide-bound PTH1R and PTH2R and discuss the potential mechanism.*

Response: We thank the referee's insightful comments. Previous study showed that the substitution of Ile5 in PTH with a His5 decreases the peptide potency on PTH2R, but the substitution of His5 in PTHrP with an Ile5 significantly increases the potency¹, which indicates that the sequence variance at residue 5 between PTH and PTHrP plays an important role in the receptor selectivity (**Supplementary Fig.12a, b**). Indeed, our structural analysis between the TIP39-bound PTH2R and the ABL-bound PTH1R showed that the substitution of Asp7 in TIP39 (the endogenous ligand of PTH2R) with the equivalent His5 in PTHrP (or ABL) may lead to a potential steric clash with the Tyr318^{5.39} in PTH2R (**Supplementary Fig.12c**). Consistent with this, previous MD simulation analysis indicated that Asp7 in TIP39 stably formed hydrogen bonds with Tyr318^{5.39}, while Ile5 in PTH made hydrophobic interactions with Tyr318^{5.39}, however, no hydrogen bond or hydrophobic interaction between His5 of PTHrP and Tyr318^{5.39} were observed² (**Supplementary Fig.12c**). These observations may explain why PTHrP has a weak action on PTH2R. We have added these descriptions in our revised manuscript (line 302-305). The figure shown below is incorporated into the Supplementary Figure.

Supplementary Fig.12 | Role of the sequence variance at residue 5 between PTH and PTHrP in the recognition of PTH1R and PTH2R.

a, Sequence alignment of the PTHrP, ABL and TIP39. b, Structural comparisons of the ABL-bound PTH1R with TIP39-bound PTH2R (PDB 7F16). c, The different residue 5 between PTH and PTHrP makes differential interactions with PTH2R, possibly explaining the weak actions between PTHrP and PTH2R.

(3) In Figure 3c, the dash lines between the receptor and ligand are shown in two colors. The author should add clear descriptions for them in the figure legend.

Response: We thank the referee for pointing out the problem. The red dash lines represent hydrogen-bond interactions and the blue dash lines represent salt bridge interactions. We have added these descriptions in our revised manuscript.

(4) TMG in Supplementary Table.1, Supplementary Figs. 2b and 3b should be expanded in full name when firstly mentioned.

Response: We thank the referee for the careful reviewing. TMG indicates the transmembrane domains of the receptor with the G protein. We have added the corresponding descriptions in our revised manuscript (Line 533-534 in Methods).

(5) The version of computational programs including RELION, cryoSPARC and Rosetta should be included.

Response: We thank the referee for pointing out the problem. We have added the version number of these programs including RELION 3.1, cryoSPARC v3.2 and Rosetta 2019.35 in the revised Methods (Line 526-557).

Reviewer #2

In the manuscript titled “Molecular insights into the distinct signaling duration for the peptide-induced PTH1R activation”, the authors present two cryo-electron microscopy structures of PTH1R bound by Teriparatide (PTH) or Abaloparatide (ABL) as well as comprehensive functional studies to reveal the underlying mechanisms for peptides recognition and their mediated prolonged or transient signaling. The work is of high quality and the article is written well. It has the potential to add to the area of GPCR pharmacology, especially for class B1 GPCRs.

Response: We are grateful to the reviewer’s positive assessment on the novelty and quality of our study.

There are few issues that should be addressed by the authors, and there are errors that must be corrected.

Major points

1. As demonstrated in the paper, PTH and PTHrP are two endogenous agonists that activating highly similar downstream signaling, but differ significantly in stabilizing distinct PTH1R states and signaling duration. The molecular determinations of the peptide-mediated signaling dynamics are of critical importance for PTH1R-targeted drug discovery, and this manuscript present significant efforts on this direction. Is there some technological problems on determining PTHrP-bound structures? Alternatively, further molecular dynamic simulation of PTHrP-bound PTH1R based on ABL-PTH1R structure could strengthen current efforts. Where, I’m not surprising to see the sequence variance at the peptide is involved in their distinct signaling duration.

Response: We thank the referee’s insightful suggestion. Indeed, ABL and PTHrP are close homologues and share high sequence identity (76%), and thereby present similar structural folding and signaling properties (see below, Fig. a-c). Thus, although our preliminary pull-down assay showed that both ABL and PTHrP could activate the receptor and form the complex with Gs protein (see below, Fig. d), we believed that determining the cryo-EM structure of the clinical drug ABL-bound complex could provide similar even more significant insights as PTHrP did. Not surprisingly and consistent with our speculations, the structures of the two peptide-bound complexes are highly similar (see below, Fig. e). Structural comparisons of the ABL-bound PTH1R with the just reported PTHrP-bound complex showed the two peptides adopted highly similar binding mode with PTH1R³. Therefore, we did not further perform the MD simulation analysis of the PTHrP-bound PTH1R using the ABL-PTH1R structure.

Functional and structural analyses between the ABL and PTHrP.

a, Sequence alignment of the PTHrP and ABL. b-c, Dose-response curves shows the signaling profiles in cAMP accumulation (b) and duration of ligand-induced cAMP signaling responses (c). d, Pull-down assay of the PTH-, PTHrP- and ABL-bound PTH1R-Gs complex. e, Structural comparisons of the ABL- and PTHrP-bound PTH1R (PDB 7VVJ).

2. The hypothesis of peptide-binding affinity in different receptor states (R_0 and R_G) that may correlate with the long/short acting and/or prolonged/transient signaling is very interesting and impressive. Such hypothesis could be further supported by the peptide-binding assay in R_0 or R_G states as well as the G protein binding. Is there any evidence or experimental data (especially for the new-designed peptide mutants) could be performed? If not, further discussion is encouraged.

Response: We thank this reviewer for the helpful comments and suggestions. Extensive previous studies had showed that the duration of cAMP signaling was highly

correlates with the affinity to the R₀ receptor conformation⁴⁻⁸. The role of residue 5 and “M” substitutions was also confirmed by binding assay in previous studies^{6,9}. Therefore, we only performed “wash out” assay to analyze the peptide-induced signaling duration of PTH1R. We totally agree with referee’s valuable suggestion. We have revised the manuscript by adding the description to the discussion section as follows as the limits of our study: “However, further investigations are required to depict a more comprehensive understanding of the molecular mechanisms underlining the differential signaling durations induced by distinct peptide agonists, including the structure determination of PTH- and PTHrP-bound PTH1R in R₀ state and extensive peptide binding assays with both the R₀ and R_G state receptor” (Line 398-403).

Minor points

1. *The orders of residue positions (1, 10, 11, 5, 8, 14) in Fig. 3c is kind of unusual, and the addition of LA-PTH in such binding pose comparison could provide further information. Besides, as there are three cryo-EM structures of LA-PTH-PTH1R-Gs complexes, the detailed PDB code should be mentioned.*

Response: We thank the referee’s valuable suggestion. The N-terminal residues (residues 1-14) in PTH and ABL play important role in receptor activation. Sequence and structure analysis showed residue divergences at positions 1, 10 and 11 of PTH and ABL showed no obvious difference to the detailed interactions. Nevertheless, we noticed obvious differences occurred at positions 5, 8 and 14. Hence, we put the similar interactions on the left (positions 1, 10, 11), and the differential interactions on the right (positions 5, 8, 14) in Fig. 3c.

LA-PTH is a PTH/PTHrP hybrid analog, with a combination of six residue substitutions (Ala^{1,3,12}, Gln¹⁰, Arg¹¹, Trp¹⁴, designated as “M” substitutions) in the N-terminal. In addition, the underlying mechanism for the more potent signaling duration of LA-PTH is resulted from multiple factors including the residue Ile5, the “M” substitutions, the C-terminal part of PTHrP and the C-terminal three residue substitutions. Therefore, comparison of the detailed interactions between three peptides with PTH1R is complicated, and we discussed the detailed interaction and underlying mechanism of LA-PTH in the Figure 4e for clarity.

Three structures of LA-PTH-PTH1R-Gs complexes with different ligand conformational states have been reported. In this study, we chose the representative full active state structure (state 1, PDB code: 6NBF) of PTH1R for structure analysis. We have added the PDB code in our revised manuscript.

2. *Fig.4 legend: Position 5 and 11 \diamond positions 5 and 11.*

Response: We thank the referee for pointing out the mistake. We have revised it accordingly.

3. *Supplementary Fig.5a. The colors of ABL- and PTH-bound PTH1R are close.*

Response: We thank the referee’s helpful suggestion. We have revised Supplementary Fig.5 for better presentation as shown below:

Supplementary Fig.5 | Structural comparisons of the ABL and PTH-bound PTH1R-Gs complexes.

a, The ABL- and PTH-bound PTH1R-Gs complexes adopt highly similar global conformations. Structures were aligned by the TMD of PTH1R. b-f, Conserved interactions between PTH1R and Gs in the ABL- and PTH-bound complexes.

4. *The source of the ligands (ABL and PTH) should be indicated and the biophysical analyses and purity assessment of the peptides should be reported.*

Response: We thank the referee's helpful comments. All the peptide ligands used in this study were synthesized in Synpeptide (Nanjing, China). We listed all the peptide ligands used in this study in the **Source data Table.1**. In addition, the biophysical analyses and purity assessment of the peptides are also provided in **source data**. We have added the descriptions in our revised manuscript (Line 636-639 in Methods).

Source data Table.1 | Sequence of the peptide ligands used in this study.

Name	Sequence
PTH	H-SVSEIQLMHNLGKHLNSMERVEWLRKKLQDVHNF-OH
PTHrP(1-36)	H-AVSEHQLLHDKGKSIQDLRRRFFLHHLIAEIHTAEI-NH2
ABL	H-AVSEHQLLHDKGKSIQDLRRRELLEKLL-{Aib}-KLHTA-NH2
LA-PTH	H-AVAEIQLMHQRAKWIQDARRRAFLHKLIAEIHTAEI-COOH
I5H-LA-PTH	H-AVAEHQLMHQRAKWIQDARRRAFLHKLIAEIHTAEI
LA-PTH (I5H-PTHrP_CT)	H-AVAEHQLMHQRAKWIQDLRRRFFLHHLIAEIHTAEI-NH2
LA-PTH (I5H-PTH_CT)	H-AVAEHQLMHQRAKWLNSMERVEWLRKKLQDVHNF-OH
LA-PTH (I5H-PTH_NT)	H-SVSEHQLMHNLGKHIQDARRRAFLHKLIAEIHTAEI-COOH

LA-PTH (15H-PTH_1)	H-SVAEHQLMHQRAKWIQDARRRAFLHKLIAEIHTAEI-COOH
LA-PTH (15H-PTH_3)	H-AVSEHQLMHQRAKWIQDARRRAFLHKLIAEIHTAEI-COOH
LA-PTH (15H-PTH_10)	H-AVAEHQLMHNRAKWIQDARRRAFLHKLIAEIHTAEI-COOH
LA-PTH (15H-PTH_11)	H-AVAEHQLMHQLAKWIQDARRRAFLHKLIAEIHTAEI-COOH
LA-PTH (15H-PTH_12)	H-AVAEHQLMHQRGKWIQDARRRAFLHKLIAEIHTAEI-COOH
LA-PTH (15H-PTH_14)	H-AVAEHQLMHQRAKHIQDARRRAFLHKLIAEIHTAEI-COOH
I5H-LA-PTH	H-AVAEHQLMHQRAKWIQDARRRAFLHKLIAEIHTAEI
M8L-LA-PTH	H-AVAEIQLLHQRAKWIQDARRRAFLHKLIAEIHTAEI
W14S-LA-PTH	H-AVAEIQLMHQRAKSIQDARRRAFLHKLIAEIHTAEI
H51-ABL	H-AVSEIQLLHDKGKSIQDLRRRELLEKLL-{Aib}-KLHTA-NH2
I5H-TER	H-SVSEHQLMHNLGKHLNSMERVEWLRKKLQDVHNF-OH

References

- 1 Gardella, T. J., Luck, M. D., Jensen, G. S., Usdin, T. B. & Juppner, H. Converting parathyroid hormone-related peptide (PTHrP) into a potent PTH-2 receptor agonist. *J Biol Chem* **271**, 19888-19893, doi:10.1074/jbc.271.33.19888 (1996).
- 2 Wang, X. *et al.* Molecular insights into differentiated ligand recognition of the human parathyroid hormone receptor 2. *Proc Natl Acad Sci U S A* **118**, doi:10.1073/pnas.2101279118 (2021).
- 3 Kobayashi, K. *et al.* Endogenous ligand recognition and structural transition of a human PTH receptor. *Mol Cell*, doi:10.1016/j.molcel.2022.07.003 (2022).
- 4 Hattersley, G., Dean, T., Corbin, B. A., Bahar, H. & Gardella, T. J. Binding Selectivity of Abaloparatide for PTH-Type-1-Receptor Conformations and Effects on Downstream Signaling. *Endocrinology* **157**, 141-149, doi:10.1210/en.2015-1726 (2016).
- 5 Noda, H. *et al.* Optimization of PTH/PTHrP Hybrid Peptides to Derive a Long-Acting PTH Analog (LA-PTH). *JBMR Plus* **4**, e10367, doi:10.1002/jbm4.10367 (2020).
- 6 Dean, T., Vilardaga, J. P., Potts, J. T., Jr. & Gardella, T. J. Altered selectivity of parathyroid hormone (PTH) and PTH-related protein (PTHrP) for distinct conformations of the PTH/PTHrP receptor. *Mol Endocrinol* **22**, 156-166, doi:10.1210/me.2007-0274 (2008).
- 7 Daley, E. J. *et al.* Ligand-Dependent Effects of Methionine-8 Oxidation in Parathyroid Hormone Peptide Analogues. *Endocrinology* **162**, doi:10.1210/endocr/bqaa216 (2021).
- 8 Sutkeviciute, I. *et al.* Precise druggability of the PTH type 1 receptor. *Nat Chem Biol* **18**, 272-280, doi:10.1038/s41589-021-00929-w (2022).

- 9 Okazaki, M. *et al.* Prolonged signaling at the parathyroid hormone receptor by peptide ligands targeted to a specific receptor conformation. *Proc Natl Acad Sci U S A* **105**, 16525-16530, doi:10.1073/pnas.0808750105 (2008).

Reviewers' Comments:

Reviewer #1:

Remarks to the Author:

In the revised manuscript, the authors really did a great job addressing most of the comments and suggestions raised by different reviewers. These additional data strengthened the conclusions of the manuscript, and the reviewer's comments have been addressed to my satisfaction. Principally, I recommend the acceptance after a couple of minor edits:

1. Supplementary Fig. 2, 3  Supplementary Figs. 2, 3
2. Figure 3 legend: positions 1, 10, 11  positions 1, 10 and 11

Reviewer #2:

Remarks to the Author:

The authors have full addressed all my questions and concerns. The manuscript is well organized and written, and I suggest its acceptance in Nature Communications.

Response to referees:

Reviewer#1:

In the revised manuscript, the authors really did a great job addressing most of the comments and suggestions raised by different reviewers. These additional data strengthened the conclusions of the manuscript, and the reviewer's comments have been addressed to my satisfaction. Principally, I recommend the acceptance after a couple of minor edits:

- 1. Supplementary Fig. 2, 3  Supplementary Figs. 2, 3*
- 2. Figure 3 legend: positions 1, 10, 11  positions 1, 10 and 11*

Response: We appreciated the referee's concerns and valuable suggestions. We have revised it accordingly.

Reviewer #2:

The authors have full addressed all my questions and concerns. The manuscript is well organized and written, and I suggest its acceptance in Nature Communications.

Response: We thank again for the referee's questions and concerns.